# Prediction of Deformation-Induced Martensite Start Temperature by Convolutional Neural Network with Dual Mode Features

**DOI:** 10.3390/ma15103495

**Published:** 2022-05-13

**Authors:** Chenchong Wang, Da Ren, Yong Li, Xu Wang, Wei Xu

**Affiliations:** 1State Key Laboratory of Rolling and Automation, Northeastern University, Shenyang 110819, China; neurenda@stumail.neu.edu.cn (D.R.); neuliyong@stumail.neu.edu.cn (Y.L.); xuwei@ral.neu.edu.cn (W.X.); 2School of Mechanical Engineering, Liaoning Petrochemical University, Fushun 113001, China; wx1979875@hotmail.com

**Keywords:** steels, microstructure, deformation-induced martensite transformation, dual mode data, deep learning

## Abstract

Various models were established for deformation-induced martensite start temperature prediction over decades. However, most of them are empirical or considering limited factors. In this research, a dual mode database for medium Mn steels was established and a convolutional neural network model, which considered all composition, critical processing information and microstructure images as inputs, was built for $M_{s}^{\sigma }$ prediction. By comprehensively considering composition, processing and microstructure factors, this model was more rational and much more accurate than traditional thermodynamic models. Also, by the full use of images information, this model has stronger ability to overcome overfitting compared with various traditional machine learning models. This framework provides inspiration for the similar data analysis issues with small sample datasets but different data modes in the field of materials science.

## 1. Introduction

Metastable austenite tailoring is a long-standing hot topic in the field of steel materials [1,2,3,4,5,6]. By transformation-induced plasticity (TRIP) in metastable austenite, different kinds of advanced steels [7,8,9], including 3rd generation ultra-high strength automobile steel, high Co-Ni secondary hardening steels [10,11], cryogenic steels [12,13], etc., obtained excellent comprehensive mechanical properties. In order to provide precise guidance for metastable austenite tailoring, various researches paid much attention to the prediction of martensitic transformation from metastable austenite including both martensite start temperature and kinetics [14].

It is widely accepted that martensitic transformation could be divided into two types: temperature-induced and deformation-induced martensitic transformation. For the prediction of temperature-induced martensite start-temperature (M_s_), various models were established including empirical formula [14], thermodynamic models [15,16,17,18,19] and machine learning strategies [20]. However, for that of deformation-induced martensite, the accumulation of previous researches was relatively insufficient. Differing from single temperature field for M_s_, deformation-induced martensite start temperature (M_d_) was reported to be affected by the coupling of both temperature and stress field [21,22]. So, the complex relationship between different loading condition and the stress/strain distribution in the material greatly limit the accuracy and stability of M_d_ predicting or testing. In order to systematically consider the coupling of temperature and stress field, the Olson-Cohen model was established by detailed dividing of different loading conditions and building the expression of the out-field related driving force by different loading conditions separately. So, this model could help to predict M_d_ with specific loading condition, which was named $M_{s}^{\sigma }$ [22,23,24,25]. Although the Olson-Cohen model provided a preliminary attempt for $M_{s}^{\sigma }$ prediction, it was still a constitutive model based on phase transformation mechanism. The complex and controversial mechanism of deformation-induced martensitic transformation greatly inhibited the further improvement of rationality and accuracy of the Olson-Cohen model. For example, the contribution of stress on driving force was simply expressed by Molar’s method, which was much different with the complex stress distribution in materials. Also, the Olson-Cohen model could not consider microstructure factors like grain size or morphology. Then, in order to make it more rational, several researchers made modifications to add the effect of grain size. In 2004, S. Takaki et al. [26] added the effect of grain size into the Olson-Cohen model by modifying the elastic strain energy in the resistance term of martensitic transformation based on the theory of lattice mismatch. This modified model was well verified in Fe-Cr-Ni ternary alloy system. In 2017, S.M.C. van Bohemen et al. [27] also added the effect of grain size into the Olson-Cohen model by adding Hall-Petch energy term to the martensitic transformation resistance term based on Hall-Petch strengthening theory. This modified model was fully verified in a relatively wider Fe-C-Mn-Si-Cr-Ni-Mo seven-element system. Although great effort was made for the improvement of the Olson-Cohen model, other microstructure factors except for grain size still could not be fully considered because their effect mechanism were still unclear and some factors like morphology could hardly be quantitatively expressed. In addition, recently, some previous research already began to build an M_s_ predictor by machine learning methods. For example, M. Rahaman et al. [20] trained various statistical learning models based on the Materials Algorithm Project (MAP) database and found an optimal model for M_s_ prediction by comparing the performance of different statistical learning strategies. However, few studies reported the application of deep learning on M_s_ or M_d_ prediction.

In order to overcome the limit of the complex mechanism of deformation-induced martensitic transformation and obtain an accurate and rational prediction model of $M_{s}^{\sigma }$, a deep learning model based on the convolutional neural network (CNN) strategy [28] was established. In this deep learning framework, all composition, loading stress and microstructure images data was used as inputs to fully reflect different factors of $M_{s}^{\sigma }$. And the advantages of this model were verified by the comparison with traditional the Olson-Cohen model and various traditional machine learning models.

## 2. Materials and Methods

### 2.1. Materials

In this research, a medium manganese steel database with different composition and process was established. Different from the traditional database for martensite start temperature (M_s_) or $M_{s}^{\sigma }$ prediction, which only contains numerical data as the value of element content or processing parameters, the database established in this research also added microstructure images labeled to every sample. Therefore, it is a dual mode database with integrated information of composition, processing and microstructure.

The chemical composition of the test steel used was 0.2 C, 3–6 Mn, 1.6 Si (in wt.%), the balance being Fe. For the preprocessing, the ingots were prepared in a vacuum induction furnace. The infrared carbon sulfur analyzer, spectrophotometer and inductively coupled plasma emission spectrometer were used to test element contents carefully. The ingots were homogenized at 1200 °C for 5 h, and then forged to the size of 120 mm × 150 mm. After forging, the alloys were hot rolled through 7 passes of hot rolling process and finally water quenched to room temperature. For heat treatment process, the alloys were normalized at 900 °C for 600 s. Further annealing was performed, specifically, alloys were first reheated to 735~790 °C for 0.5~15 min based on the composition difference. Then alloys were cooled at a cooling rate of 10 °C/s while applying a compressive force of 1000 or 2000 N during cooling. The final heat treatment process is shown in Figure 1.

For microstructure, ZEISS Gemini SEM 300 scanning electron microscope was used to get 468 microstructure images for 38 samples, with 1024 × 768 pixels. All the samples for obtaining the micrographs used for this model were taken from the rolled plate along the rolling direction. All the samples were standardized polished by the automatic polishing machine with exactly the same run parameters. Then, all the observation samples were etched with 4% Nital solution in 10 s. For labels, the DIL805AD deformation thermal expansion instrument was used for $M_{s}^{\sigma }$ testing. The compression module of the DiL805AD thermal dilatometer was used for applying the compressive force on the $M_{s}^{\sigma }$ testing samples. The size of all the $M_{s}^{\sigma }$ testing samples was Φ5 × 10 mm. Before the testing, all the dilatometer samples were cleaned by ultrasonic to improve the surface cleanliness. Finally, the standard tangent method was applied to get the value of $M_{s}^{\sigma }$ temperature from the dilation curves. So far, the dual mode database with integrated information of composition, processing and microstructure were established. The composition and the microstructure images of the samples in the database were attached as the database file.

### 2.2. Details of the CNN Model

Based on the dual mode database, the CNN model was established and trained for $M_{s}^{\sigma }$ prediction. The framework is shown in Figure 2. Before training, data preprocessing was used for data augmentation. The microstructure images obtained by a scanning electron microscope (SEM) were firstly cut to 336 × 336 sub-images. Then, further data augmentation was made by turning or mirroring. After data preprocessing, the sub-images were used for training the parameters in the convolutional and pooling layers in the CNN model. Also, dropout strategy with the rate of 0.6 was used to reduce the risk of overfitting for a deep learning network. Different from traditional CNN models using for image classification or recognition, not only image data, but also numerical data like composition and processing parameters were used as inputs of the network. After training the convolutional and pooling layers by sub-images, numerical data including the content of C, Mn, Si, intercritical annealing temperature (T), intercritical annealing time (t), loading stress for $M_{s}^{\sigma }$ testing (F) were all introduced to the fully connected (FC) layer of the CNN architecture with splicing neurons. The ratio of the neuron amount for image data and numerical data was set to 1:1. Therefore, the parameters for the FC layer were trained by all microstructure, composition and critical processing information and finally the value of $M_{s}^{\sigma }$ was predicted.

For the CNN model, the choice of the CNN architecture is greatly beneficial for accurate prediction. After structure and parameter optimization, six convolutional layers with the filter size of 3 × 3 and three pooling layers were used to extract microstructure images information. Further, the composition and processing information are introduced from the fully connected layer by means of neuron splicing. Two fully connected layers with 1024 neurons were set finally, containing comprehensive information. The Adam algorithm was chosen as the optimizer and the learning rate was set as 0.0001.

The squared correlation coefficient (*R*^2^) and mean absolute error (MAE) were adopted to evaluate the generalization ability of the CNN models. The calculation methods are given by Equations (1) and (2):(1)$$R^{2}=\frac{{\left(n\sum_{i=1}^{n}f(x_{i})y_{i}-\sum_{i=1}^{n}f(x_{i})\sum_{i=1}^{n}y_{i}\right)}^{2}}{\left(n\sum_{i=1}^{n}f{\left(x_{i}\right)}^{2}-{\left(\sum_{i=1}^{n}f(x_{i})\right)}^{2})-(n\sum_{i=1}^{n}y_{i}{}^{2}-{\left(\sum_{i=1}^{n}y_{i}\right)}^{2}\right)}$$
(2)$$MAE=\frac{1}{n}\overset{n}{\underset{i=1}{\sum }}\left|f(x_{i})-y_{i}\right|$$
where n is the number of samples and $f(x_{i})$ and $y_{i}\;$represent the predicted and experimental values of the ith samples, respectively. All results in this article were generated using the Python deep learning framework Keras.

### 2.3. Details of the Olson-Cohen Model

In this research, the $M_{s}^{\sigma }$ temperature was also calculated using the Olson-Cohen model [22,23,29] for comparison, which is presented as follows:(3)$$\Delta G^{Chem}(T)+\Delta G^{Mech}=-g_{n}-W_{f}\;(T)$$
where $\Delta G^{Chem}$ and $\Delta G^{Mech}$ are chemical and mechanical driving force of martensitic transformation; $g_{n}$ is a constant, which includes the strain and interfacial energies and defect size. $W_{f}$ is the frictional work of interface motion. Both $\Delta G^{Chem}$ and $W_{f}$ depended on temperature. Thus, critical temperature, which was equal to $M_{s}^{\sigma }$ temperature, could be obtained by solving Equation (3).

#### 2.3.1. Chemical Driving Force

The chemical driving force $\Delta G^{Chem}(T)$ is the Gibbs free energy difference between the face-centered cubic (FCC) and body-centered cubic (BCC) phases. It is directly calculated using Thermo-Calc software. The value of the *T* dependent parameters for calculating the chemical driving force is directly obtained from the TCFE9 database in the Thermo-Calc software.

#### 2.3.2. Mechanical Driving Force

The mechanical work per unit volume done by an applied stress, which assisted the martensitic transformation, could be expressed by Equation (4) [30].
(4)$$\Delta G^{Mech}=\tau \gamma_{0}+\sigma_{0}\varepsilon_{0}$$
where $\gamma_{0}$ and $\varepsilon_{0}$ are the resolved shear and normal strains, respectively; τ and $\sigma_{0}$ are the resolved shear and normal stresses on the planes in the directions of $\gamma_{0}$ and $\varepsilon_{0}$, respectively.

After, derived by Mohr’s circle [31,32], Equation (4) could be expressed as Equation (5) for tensile uniform ductility,
(5)$$\Delta G^{Mech}=-V_{m}\sigma (\frac{\delta +\sqrt{\delta^{2}+\gamma_{0}^{2}}}{2})$$
where $V_{m}$ is the molar volume of FCC phase; σ is the mean stress; δ is the dilatation of martensitic transformation. The dilatation could be expressed by the change of lattice constant, based on Equations (6)–(8) [33].
(6)$$\alpha_{\mathrm{BCC}}(Å)=2.8664-0.014x_{c}+0.00055x_{\mathrm{Mn}}$$
(7)$$\alpha_{\mathrm{FCC}}(Å)=3.556+0.0453x_{c}+0.00095x_{\mathrm{Mn}}$$
(8)$$\delta =2{\left(\frac{\alpha_{\mathrm{BCC}}}{\alpha_{\mathrm{FCC}}}\right)}^{3}-1$$
where $x_{i}$ is in wt.% of element *i*.

#### 2.3.3. Frictional Work of Interface Motion

The frictional work of interface motion can be divided into two parts: athermal and thermal contributions, as shown in Equation (9) [22,23]:(9)$$W_{f}(T)=W_{athermal}+W_{thermal}(T)$$

The thermal and athermal contributions are expressed by Equations (10)–(12):(10)$$W_{athermal}=\sqrt{\underset{i}{\sum }K_{\mu ,\;\;i}^{2}x_{i}}+\sqrt{\underset{j}{\sum }K_{\mu ,\;\;j}^{2}x_{j}}+\sqrt{\underset{k}{\sum }K_{\mu ,\;\;k}^{2}x_{k}}$$
(11)$$W_{thermal}(T)=W_{0}{\left[1-{\left(\frac{T}{T_{\mu }}\right)}^{p}\right]}^{q}$$
(12)$$W_{0}=W_{\mathrm{Fe}}+\sqrt{\underset{i}{\sum }K_{0,\;\;i}^{2}x_{i}}+\sqrt{\underset{j}{\sum }K_{0,\;\;j}^{2}x_{j}}+\sqrt{\underset{k}{\sum }K_{0,\;\;k}^{2}x_{k}}$$
where $K_{\mu ,i}$ and $K_{0,i}$ are the athermal and thermal coefficients for element i; T is the absolute temperature; and $T_{\mu }$, which is dependent on the interfacial rate, is the critical temperature. When $T_{\mu }<T$, $W_{thermal}$ is negligible; p and q are exponential parameters; $W_{Fe}$ is the thermal contribution of Fe; i represents element C; j
*and k* represent Mn and Si, respectively.

In summary, the $M_{s}^{\sigma }$ temperature can be found by combining Equations (3)–(12), the parameters used in the final calculation are shown in Table 1.

## 3. Results

Figure 3 shows the performance of the CNN model for both training and testing set. For three times training of the samples with the ratio of 8:2, the error bars for most samples were acceptable small and most predicted values for the testing sets are basically distributed on the straight line with a slope of 1, illustrating that the model shows high prediction accuracy and stability. It is also clear that the mean value of R^2^ and MAE for the testing set were 97.9% (±1.1%) and 2.3 °C (±0.5 °C) respectively, which is basically similar with the performance for training set (98.1% (±1.0%) for R^2^ and 2.2 °C (±0.4 °C) for MAE). For the dual mode database used in this research, only 38 samples were fabricated and treated for $M_{s}^{\sigma }$ testing, which is a typical small sample problem. It is extremely difficult to directly build a stable artificial intelligence (AI) model without overfitting. However, with adding image data, more information was provided for every sample, which can help to overcome the risk of overfitting by information enhancement. Also, image data is easy to be augmented by cutting, turning, mirroring, etc. as mentioned in Section 2.2. Therefore, the proposed CNN model provided a useful method for reducing the cost and time consuming for sample fabricating during establishing a database by making full use of dual mode data.

## 4. Discussion

### 4.1. Comparison of CNN Model and Olson-Cohen Model

In order to further explain the advantages of the proposed CNN model, the comparison with the traditional Olson-Cohen model [22,23,24,25] was made. For the Olson-Cohen model, $M_{s}^{\sigma }$ is calculated by the law of energy conservation based on thermodynamic theory. The comparison results are shown in Figure 4, in which the accuracy improvement of the proposed CNN model is clear. The MAE of the Olson-Cohen model is 33.5 °C, which is ~30 °C higher than the proposed CNN model. These results are reasonable because the the Olson-Cohen model is an ideal model with various assumptions as most thermodynamic models. Although the Olson-Cohen model was widely used for decades and helped to successfully design several kinds of high performance steels [25], some limits still exist and need to be modified. Firstly, as a model based on equilibrium thermodynamics, the effect of processing, like intercritical annealing temperature or time, is not considered in this model. However, processing can significantly affect the constitution and morphology of the microstructure, which is critical for $M_{s}^{\sigma }$. Also, for the Olson-Cohen model, the contribution of the mechanical driving force is simply estimated by empirical equation. However, mechanical driving force is a complex term also highly related with microstructure. Empirical equation without considering microstructure factors is probably not available to reflect its contribution precisely. Therefore, it can be seen that, without considering microstructure or processing factors, all the samples with the same composition and loading stress for $M_{s}^{\sigma }$ testing has the same predicted value by the Olson-Cohen model. This obvious error makes the accuracy of the Olson-Cohen model significantly lower than the proposed CNN model, which considers microstructure factors by image data.

### 4.2. Comparison with Different Machine Learning Methods

In order to further explain the advantages of the proposed CNN model compared with traditional machine learning methods, various other machine learning strategies, including support vector regression (SVR), XGBoost (XGB), random forest (RF), gradient boosting regression (GBR) and Adaboost (ADB) were also trained by the same database used in this research. However, because these strategies were regression methods simply used to process numerical data, the image information in this database was not used for training these models. Figure 5 clearly showed that, compared with the proposed CNN model, all the other models had lower R^2^, higher MAE for the testing set and larger error bars. This means that all the other models have a much stronger trend of overfitting and instability than the proposed CNN model. Usually in various small sample problems, SVR is an optimal choice for regression. However, for the $M_{s}^{\sigma }$ prediction in this research, it surprisingly showed relatively worse performance than other strategies. It clearly showed that the intrinsic relationship between composition, processing and $M_{s}^{\sigma }$ is more complex than many other traditional small sample problems and it is far beyond SVR’s ultimate regression ability. For other ensemble learning algorithms, which are more powerful for regression, although they have the ability to achieve more complex regression, more data are also needed for their training. Also, as methods for numerical data, these ensemble learning algorithms can hardly use image information for data enhancement. Therefore, it is also understandable that insufficient training data leads to overfitting of these ensemble learning models. On the contrary, by using the image information as data enhancement, the proposed CNN model solved the complex problem of $M_{s}^{\sigma }$ prediction within the limit of small sample database.

### 4.3. Analysis of Different Model Parameters

In order to obtain the optimal architecture of the proposed CNN model. The CNN models with different ratio of the neuron amount for image data and numerical data were systematically built and trained in the range of 1:7 to 7:1. The comparison results are shown in Figure 6. The results of both R^2^ (Figure 6a) and MAE (Figure 6b) clearly showed that 1:1 is the best ratio to obtain the optimal performance. It also indicates that microstructure and composition/processing parameters have nearly the same importance for $M_{s}^{\sigma }$ prediction, which further proved the rationality of introducing both image and numerical data in this proposed CNN model. It could also be clearly shown that nearly all the R^2^ of the CNN models with different ratio of the neuron amount for image data and numerical data are higher than 0.9, except for an extreme division (7:1). This means that the performance of the model is not extremely sensitive to the ratio of the neuron amount for image data and numerical data, which further proves its robustness and stability.

## 5. Conclusions

A dual mode database with both composition/processing parameters and microstructure images was established in the system of medium Mn steels. Based on the database, a convolutional neural network model considering composition, critical processing and microstructure factors was built for $M_{s}^{\sigma }$ prediction. Compared with the traditional Olson-Cohen model, which does not consider microstructure or processing factors, this model is more rational and accurate because microstructure and composition/processing parameters have nearly the same importance for $M_{s}^{\sigma }$ prediction. Compared with various traditional machine learning models, this proposed model also shows stronger ability of avoiding overfitting. Also, the idea of making full use of dual mode data by CNN architecture can help to reduce the cost and time consumed for sample fabricating while establishing a database. It is beneficial for solving various similar small sample problems in the field of materials science.

## Figures and Tables

**Figure 1 materials-15-03495-f001:**
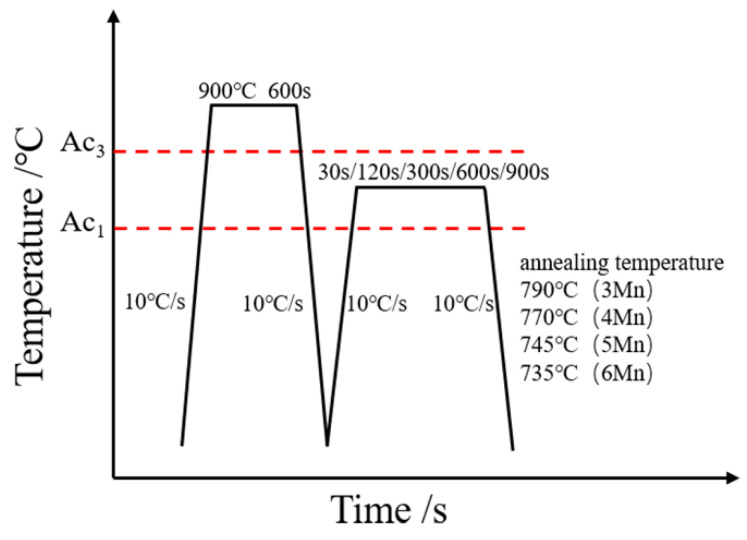
The final heat treatment process for the medium manganese steel samples. (Ac1 and Ac3 represents the initial and final temperature of austenite transformation during heating, respectively).

**Figure 2 materials-15-03495-f002:**
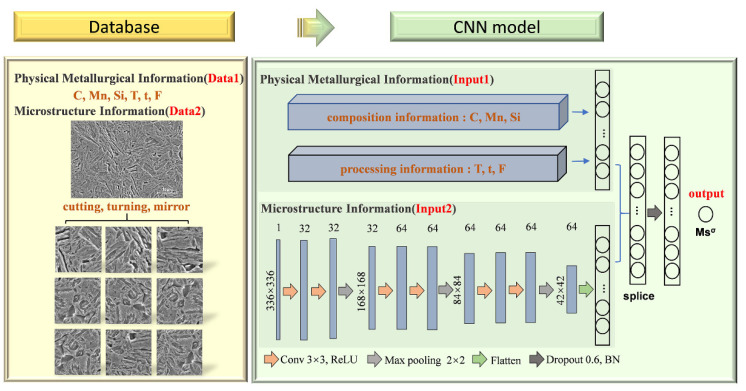
The framework of the proposed CNN model.

**Figure 3 materials-15-03495-f003:**
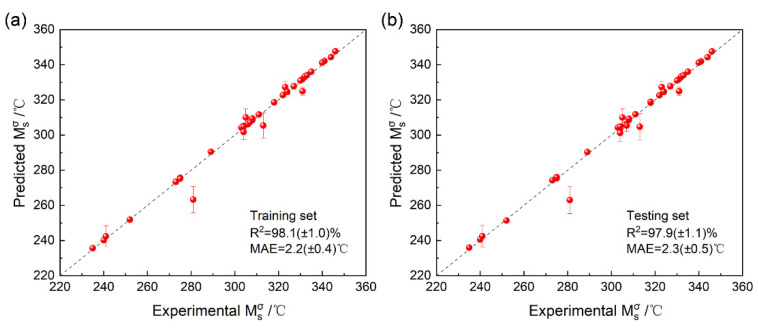
The performance of the proposed CNN model for $M_{s}^{\sigma }$ prediction: (**a**) training set; (**b**) testing set.

**Figure 4 materials-15-03495-f004:**
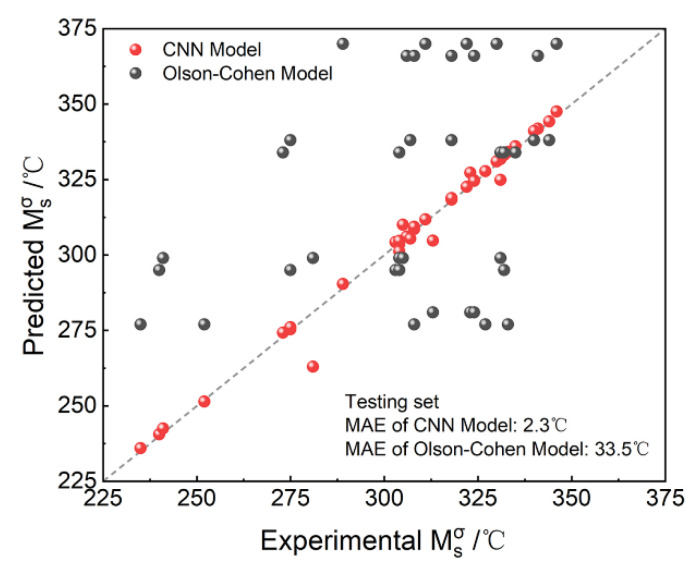
The comparison results with the Olson-Cohen model.

**Figure 5 materials-15-03495-f005:**
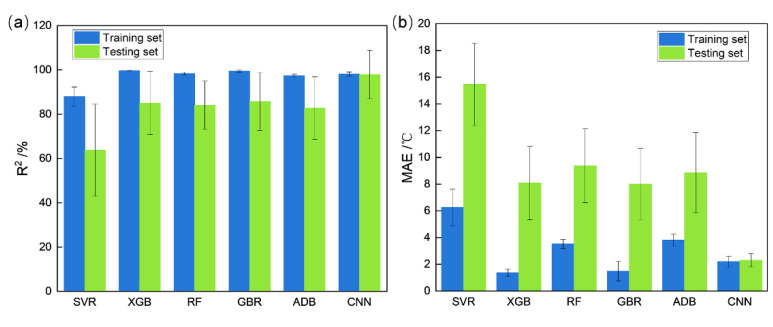
The results of different machine learning methods: (**a**) the results of R^2^; (**b**) the results of MAE.

**Figure 6 materials-15-03495-f006:**
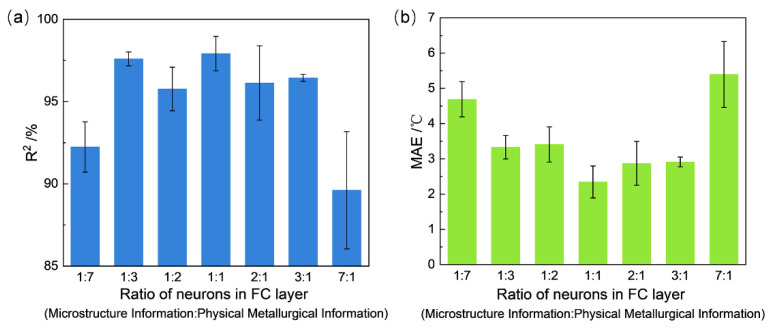
The results of the CNN models with different ratio of the neuron amount for image data and numerical data: (**a**) the results of R^2^; (**b**) the results of MAE.

**Table 1 materials-15-03495-t001:** Parameter values used to calculate the $M_{s}^{\sigma }$ temperature.

Parameter	Value	Parameter	Value
$K_{\mu ,\;C}$	4009 J/mol	$K_{0,\;\mathrm{Mn}}$	4107 J/mol
$K_{\mu ,\;\mathrm{Mn}}$	1980 J/mol	$K_{0,\;\mathrm{Si}}$	3867 J/mol
$K_{\mu ,\;\mathrm{Si}}$	1879 J/mol	$W_{\mathrm{Fe}}$	836 J/mol
$K_{0,\;C}$	21,216 J/mol	$T_{\mu }$	510 K
*p*	0.5	$\gamma_{0}$	0.13
*q*	2	$g_{n}$	750

## Data Availability

The data presented in this study are available upon request from the corresponding author.

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
