# Peer review of "Prediction of Deformation-Induced Martensite Start Temperature by Convolutional Neural Network with Dual Mode Features"

_materials, 2022, doi:10.3390/ma15103495_

Round 1

Reviewer 1 Report

This work is about an improved prediction of deformation induced martensite start temperature in steels. The subject is relevant and important: adequate taking into account the effect of the microstructure and processing parameters is well desired. The method based on the Olson-Cohen model and the calculations were implemented by a convolutional neural network model. It is concluded that the microstructure and processing parameters have nearly the same importance.

The results obtained are convincing and the overall paper provides a good impression, and I propose to accept it after minor corrections.

 Comments/questions:

  • in row 89: the force was tensile or compressive ?
  • in row 90: “was” should be replaced by “is”
  • 1: a chines symbol is present in the insert. What are meanings of parameters Ac1 and Ac3?
  • in row 101: “was” should be replaced by “is”
  • Regarding the equations to be solved for Md : in rows 165 and 166 it is written that “Md can be found by combining Eqs. (3)-(12)” But no explicit relation is given for the T dependence of the chemical driving force (and no data related to this are given in Table 1).

Optional comment: I am not sure whether using Md as the deformation induced martensite start temperature, is better than the original definition (see ref. 23) of Mss: “A reversal of the temperature dependence of the yield stress occurs at the temperature Mss".

Reviewer 2 Report

The paper addresses the study of a dual (micrographs + values of composition and stress) prediction model of Md (deformation induced martensite start temperature) on a C-Mn-Si steel.

In my opinion, the results presented in this paper are of interest to the materials science and technology community. Nevertheless, authors should address the following comments before publishing:

  1. I miss in the introduction a mention of successful applications of deep learning on metallurgy. This would better contextualise the paper.
  2. Line 38. Should not be written ‘was reported to be affected’ instead of ‘was affected’?

  3. Line 80, when describing the chemical composition of the samples under study, a hyphen should substitute the 'almost equal to' symbol before Mn.

  4. Figure 1 has some Chinese characters.

  5. Line 94 says 'For labels, 94 DIL805AD deformation thermal expansion instrument was used for Md testing'. Nevertheless, I have not found in the text the details of the methodology used for getting the Md values. These details are important to understand the methodology used, in order to extensively apply it to other alloys. Please describe or include a reference where these details are described.

  6. I find the same lack of information regarding the micrographs used for the model. Details on the sample preparation (orientation, grinding/polishing, etching...) should be reported. I consider these details rather important, due to the scope of the journal (which focuses on the Materials Science point of view).

  7. Line 125. I would include the word 'algorithm' when mentioning to Adam, in order to clarify.
  8. Line 164 says that j represents Mn and Si. So, what about k?
